# Comparison of growth models to describe growth from birth to 6 years in a Beninese cohort of children with repeated measurements

Shukrullah Ahmadi [1], Florence Bodeau-Livinec,[1,2] Roméo Zoumenou,[3] André Garcia,[4] David Courtin,[4] Jules Alao,[5] Nadine Fievet,[4] Michel Cot,[4] Achille Massougbodji,[6] Jérémie Botton [7]

For numbered affiliations see end of article.

**Correspondence to**
Shukrullah Ahmadi;
shukrullah.ahmadi@inserm.fr

## ABSTRACT

**Objective** To select a growth model that best describes individual growth trajectories of children and to present some growth characteristics of this population.

**Settings** Participants were selected from a prospective cohort conducted in three health centres (Allada, Sekou and Attogon) in a semirural region of Benin, sub-Saharan Africa.

**Participants** Children aged 0 to 6 years were recruited in a cohort study with at least two valid height and weight measurements included (n=961).

**Primary and secondary outcome measures** This study compared the goodness-of-fit of three structural growth models (Jenss-Bayley, Reed and a newly adapted version of the Gompertz growth model) on longitudinal weight and height growth data of boys and girls. The goodness-of-fit of the models was assessed using residual distribution over age and compared with the Akaike Information Criterion (AIC) and Bayesian Information Criterion (BIC). The best-fitting model allowed estimating mean weight and height growth trajectories, individual growth and growth velocities. Underweight, stunting and wasting were also estimated at age 6 years.

**Results** The three models were able to fit well both weight and height data. The Jenss-Bayley model presented the best fit for weight and height, both in boys and girls. Mean height growth trajectories were identical in shape and direction for boys and girls while the mean weight growth curve of girls fell slightly below the curve of boys after neonatal life. Finally, 35%, 27.7% and 8% of boys; and 34%, 38.4% and 4% of girls were estimated to be underweight, wasted and stunted at age 6 years, respectively.

**Conclusion** The growth parameters of the best-fitting Jenss-Bayley model can be used to describe growth trajectories and study their determinants.

## INTRODUCTION

Insufficient growth during childhood adversely affects later health outcomes. According to estimates from the Global Burden of Diseases, Injuries and Risk Factors study 2016,[1] the prevalence of stunting,

### Strengths and limitations of this study

► This is the first study to assess the comparative merits of growth models on child anthropometric data in a semirural setting in sub-Saharan Africa with a high prevalence of undernutrition.

► The generalisability of these results cannot be guaranteed because the study sample is not representative of Beninese children.

► Other models could have been tested but the overall fit of the selected models was very optimal and they present advantages for future studies in this population.

wasting and underweight (based on the definitions using the WHO 2006 growth standards) in children under 5 years were, respectively, 36.6%, 8.6% and 19.5% in sub-Saharan Africa (SSA) in 2015. Furthermore, more than 23% of under-5 mortality was attributable to child growth failure, being thus the second leading risk factor for child mortality in this region.

Mathematical growth modelling is a powerful tool for the study of child growth and growth trajectories. It consists of fitting models to physical growth data (eg, weight, length/height and head circumference) to obtain an appropriate growth curve that will conveniently summarise growth information provided by weight and height measurements of children, even from irregularly spaced growth measurements.[2] Clinicians routinely compare growth data to growth charts to identify impaired growth trajectories in children. Indeed, looking at growth trajectories (weight, height/length or body mass index) has gained importance for several purposes. It is becoming essential for surveillance to identify abnormal growth trajectories.[3] In the postnatal period, length/height and weight surveillance is an essential tool for

monitoring child growth. It is also used in research to study determinants of growth, including identification of modifiable risk factors and sensitive age periods during which interventions may be especially useful to achieve optimal growth.[4] Characterising growth trajectories is also important to study whether past growth is associated with future growth patterns or later health outcomes.[5]

There are several structural and non-structural growth models to describe child growth, among them are the Jenss-Bayley model,[6] the Reed model[278] and the Gompertz functions.[9] These three models are among the most common structural models described in the literature, while non-structural models are mainly polynomial and splines,[10–13] as cited in Chirwa *et al* (2014).[14] The parameters of structural models have a biological basis. In contrast, non-structural models do not formulate any particular form of growth curve and may demonstrate instability at extremities. Besides, the parameters obtained from non-structural models do not provide any biological interpretation.[12 15]

Many studies have analysed child growth data in Africa, but few studies[8 13 14 16–19] have applied structural or non-structural growth models on African longitudinal growth data. Very few of these studies[8 14] have assessed the comparative merits of models like the Jenss-Bayley and Gompertz on child growth in SSA. Furthermore, as to our knowledge, no study has compared the goodness-of-fit of the adapted Gompertz growth model on child growth data.

This study aimed to compare the fit of three structural growth models (the Jenss-Bayley model, the Reed model and a new model which is an adaptation of the Gompertz model) applied to weight and height growth data in a population of Beninese children aged 0 to 6 years, living in a semirural region, and to describe their growth using the selected model.

## METHODOLOGY
### Study design and data sources
Weight and height measurements were available from a prospective cohort study of Beninese children followed from birth to age 6 years. These children who participated were born within the MiPPAD (Malaria in Pregnancy Preventive Alternative Drugs) clinical trial (NCT00811421) and followed in three ancillary studies, the APEC (Anaemia in pregnancy: etiology and consequences), TOLIMMUNPAL and TOVI studies, and finally assessed in the EXPLORE study at 6 years, on average. The research objectives, design, sample size and follow-up period varied for these studies.

Participating children were born of HIV-negative mothers (n=1182) enrolled in the MiPPAD clinical trial in 2011 in the district of Allada, south Benin.[20] The MiPPAD trial, which enrolled pregnant women before 28 weeks of gestation at the first antenatal care visit, compared the efficacy of two intermittent preventive treatments for malaria in pregnancy (IPTp). Height and weight of children were

assessed at birth and women were requested to bring their infants to the health centre when the babies were aged 1 month, and also 9 and 12 months. Children included in our analyses met the following inclusion criteria: being born within the MiPPAD clinical trial, and assessed at least twice with valid anthropometric measurements (weight and height) between birth and age 6 years within MiPPAD, APEC, TOLIMMUNPAL, TOVI or EXPLORE studies.

Within this (MiPPAD) cohort, a subsample of 400 children was followed more closely. These children were followed from birth to age 2 years in the APEC[21] and TOLIMMUNPAL studies.[22] The APEC study included the first 400 offspring born within the MiPPAD study including anthropometric measurements of children at 6 and 9 months. Infants assessed in the APEC study were also followed up in the TOLIMMUNPAL study between birth and 24 months, with further anthropometric measurements at 15, 18, 21 and 24 months.

The TOVI study followed up 747 singleton 1-year-old children born within MiPPAD. Among these children, only 92 children randomly selected were assessed between 3 and 5 years for anthropometric measurements.[23] Finally, the children were assessed within the EXPLORE study at age 6 years on average for anthropometric measurements (2016 to 2018).[24 25] Online supplementary figure 1 shows participants inclusion and how different ancillary studies are embedded in this cohort.

Appropriate participant consent and ethics approval were obtained for the different studies included.

### Patient and public involvement
Patients and the public were not directly involved in the planning of this study. Results will be communicated to participants via community meetings with local leaders.

### Anthropometric measurement
Measurements were performed by trained staff at each visit, with children wearing light clothes only and no shoes. From birth up to around age 2 years, children's length was measured to the nearest 1 mm using a locally manufactured wooden measuring scale according to the WHO guidelines.[26] From birth up to age 2 years, weight was measured using an electronic baby scale (Seca type 354) with a precision of 10 g. From age 2 years onward, standing weight and height was measured for all children. Height was measured by a locally manufactured length board. The length board comprised of a measuring board mounted to the wall with an attached measuring tape and movable right angle headpiece. Weight was measured using an analogue scale (Camry DT602). At age 6 years, children were weighed with the Tanita scale.

### Data inclusion
Participants with at least two weight or height measurements anytime between birth and age 6 years were included in the analysis. Weight-for-age z-scores (WAZ) and height-for-age z-scores (HAZ) were calculated based

on the WHO growth standards.[27] To identify outliers, as the distribution of the z-scores was shifted to the left, the distribution was corrected by centring it to the mean. Then measurements with centred WAZ or HAZ greater than +4 or less than −4 were considered outliers for growth and were systematically excluded from the analysis. Additionally, individual weight and height profiles were plotted to visually check individual growth trajectories for plausibility. 65 height (out of 5291 measures) and 35 weight observations (out of 5291 measures) were excluded. They were considered as outliers on the descriptive graph even after the systematic removal of observations with WAZ or HAZ greater than +4 or less than −4. The final analyses had 961 participants (461 boys and 500 girls). The number of children with anthropometric measurements at different age intervals is presented in the online supplementary table 1.

### Growth models

The following three structural growth models were compared: the Jenss-Bayley model, the Reed model and a new model which is an adaptation of the Gompertz model. These models are presented in the equations 1, 3 and 4, where $E(y_{ij})$ is the expected value (weight or length of child $i$ at $j^{th}$ occasion, $t_{ij}$ is age in days, $A_i$, $B_i$, $C_i$ and $D_i$ are the parameters of the functions for the $i^{th}$ child.

The Jenss-Bayley model[6] is presented in equation 1 and as another parameterisation in equation 2. Briefly, a or $\exp^A$ is the predicted value at birth (t=0); b or $\exp^{-B}$ is the asymptotic slope or growth rate from about 2 years onward; d or $\exp^{-D}$ allows for decelerating growth (decreasing exponential function) during infancy. The parameter c or $\exp^C$ can reflect the degree of catch-up growth. These four parameters or model coefficients can be obtained for each individual—thanks to a mixed-effects modelling approach—and represent the global individual growth trajectory. Parameters (or a combination of them, eg, the velocity) can then be studied in association with the determinants of growth or other independent variables.[28]

$$E(y_{ij}) = a_i + b_i \cdot t_{ij} - (\exp^{c_i + d_i \cdot t_{ij}}) \quad (1)$$

$$E(y_{ij}) = \exp^{A_i} + \exp^{-B_i} \cdot t_{ij} + \exp^{C_i} \cdot (1 - \exp^{-\exp^{-D_i \cdot t_{ij}}}) \quad (2)$$

The Gompertz model[29] (equation 3 with $D_i$=0) is a three-parameter model commonly used to model human growth and development.[30 31] Parameter $A_i$ represents the upper asymptote (maximum value), $C_i$ represents growth rate and $B_i$ is related to weight at birth ($t$=0). The Gompertz model theoretically approaches a null asymptote at later ages and to better represent the positive linear growth between about 3 and 6 years, the Gompertz growth function[9 31] was expanded by adding a fourth parameter $D_i$ on height and weight modelling. This adaptation allows the model to have a non-flat asymptote (slope=velocity different from 0). This permits to fit the linear part of the growth at the later ages in our data (constant linear growth velocity). The model is non-linear in the model parameters.

$$E(y_{ij}) = A_i \cdot \exp(-B_i \cdot \exp(C_i \cdot t_{ij})) + D_i \cdot t_{ij} \quad (3)$$

In the Reed model,[7] parameter $B_i$ is the asymptotic slope or growth rate/velocity of preschool growth, $C_i$ is an important component of rapid early childhood growth[2 32] and models the exponential growth deceleration, $D_i$ is used to better model early life variations in growth, and $A_i$ is the intercept. The model is as follows:

$$E(y_{ij}) = A_i + B_i \cdot t_{ij} + C_i \cdot \ln(t_{ij} + 1) + \frac{D_i}{t_{ij}} \quad (4)$$

### Statistical methods

The Jenss-Bayley model, the Reed model and the adapted Gompertz model were applied to weight and height growth data of children from birth to age 6 years.

The Reed fixed-effects model was fit using the lm (linear model) function in R,[33] and the Jenss and Gompertz models using the nls (non-linear least squares) function in R[33] to obtain initial values more easily.[28] Then the models were fitted to weight and height data using the SAEMIX[34] package in R-3.5.1 software[33] as applied in another study.[28] Due to convergence issues with unstructured matrices, a diagonal variance-covariance matrix was selected for random effects.

Growth trajectories of the children were estimated using a recently proposed method described elsewhere.[28] Briefly, weight and height growth trajectories and growth velocities (first derivative of the equation) was estimated using the model presenting the best fit (the Jenss-Bayley model). Fixed-effect parameters (A, B, C and D) obtained from the mixed-effects model using SAEMIX package were used to represent average growth trajectories. The Jenss-Bayley model then allowed predicting individual weight and height by substituting the individual model parameters into their corresponding model equation (equation 2). Similarly, individual growth velocities were calculated by substituting determined individual parameters into the derivative of the Jenss-Bayley model. As a derivative of the Jenss-Bayley model, the following equation can be used to estimate growth velocity over time.[28]

$$\frac{dy}{dt} = \exp^{-B_i} + \exp^{C_i - D_i - \exp^{-D_i \cdot t_{ij}}} \quad (5)$$

Finally, undernutrition (underweight, stunting and wasting) of children was estimated at age 6 years using predicted data by the best-fitting model. Children with WAZ <−2 were classified as underweight, those with HAZ <−2 as stunted, and those with weight-for-height-z-score <−2 were classified as wasted.[27]

### Comparison of goodness-of-fit

The three non-nested models were compared using the Akaike Information Criterion (AIC), delta AIC and Bayesian Information Criterion (BIC). Lower AIC and BIC values indicate a better fit. The delta AIC[35] for a candidate model, is the difference between the AIC values of the best and the candidate models. This difference (delta AIC) is then used to determine the amount of support for each candidate model. If the delta AIC is <2, there is

**Table 1** Number of anthropometric measures per child and sex

| Gender | N | Growth dimension | Measures | | | | | |
|--------|---|------------------|----------|-----|-----|--------|-----|-----|
| | | | N | Min | Q1 | Median | Q3 | Max |
| Girls | 500 | Weight | 2680 | 2 | 4 | 6 | 9 | 11 |
| | 500 | Length/height | 2680 | 2 | 4 | 6 | 9 | 11 |
| Boys | 461 | Weight | 2514 | 2 | 5 | 6 | 9 | 11 |
| | 461 | Length/height | 2514 | 2 | 5 | 6 | 9 | 11 |

Max, maximum; Min, minimum; N, total number of measurements; Q1, first quartile; Q3, third quartile.

substantial evidence to support the candidate model (ie, the candidate model is almost as best performing as the best model). If the delta AIC is between 4 and 7, there is considerably less support for the candidate model to be the best model. Finally, if the delta AIC is >10, there is essentially no support for the candidate model to be better than the best model. Additionally, residuals of the models were estimated by subtracting observed measurements (weight and height) from the model-predicted ones. The distribution of the residuals of the models was plotted against age. The residual SD (RSD) for each model was also reported.

## RESULTS

Three models were fitted to weight and height growth data of 961 children from birth to age 6 years separately for boys and girls (table 1). Children had between two and 11 weight and height measurements (table 1), and 75% of girls had at least four measurements (five in boys). Some descriptive characteristics of the study population are presented in online supplementary table 2. The overall mean maternal age at age 1 year of the study child was 25.8 years (SD 5.6). Almost three-fourth of all mothers could not read or write at the start of the cohort. Almost 10% of mothers were underweight before their pregnancy. Almost 10% of boys and 14% of girls had a lower birth weight (<2.5 kg) and less than 7% of children were born preterm. Almost 10% of children had malaria at 1 year of age and almost 74% of boys and 67% of girls had anaemia at 1 year of age.

The distribution of the residuals over age for weight and height of both boys and girls showed that, overall, residuals were centred around zero for all ages and there was no strong dependence of residual distribution over ages for all the models (figures 1 and 2).

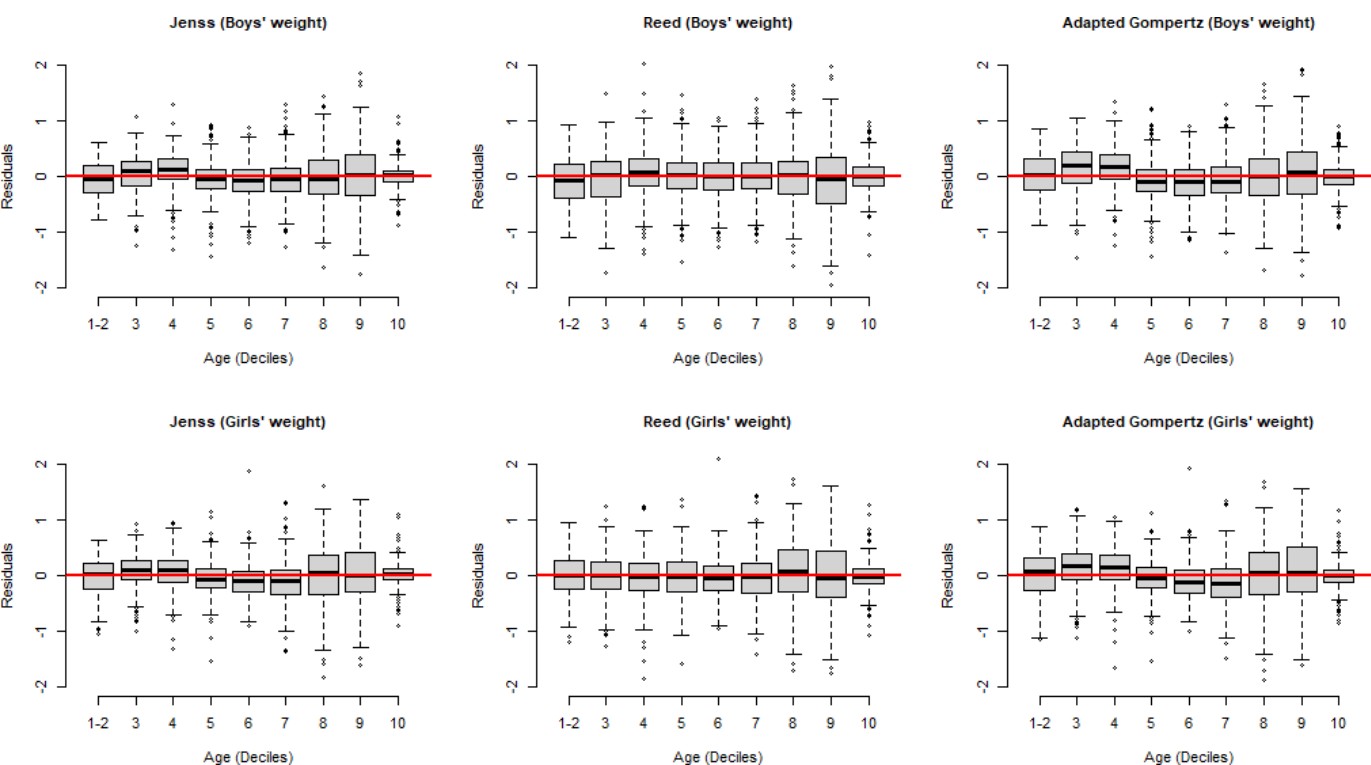

**Figure 1** Residuals of weight models for boys and girls (data of the first two deciles was gathered, as there were more than 10% of the data at birth).

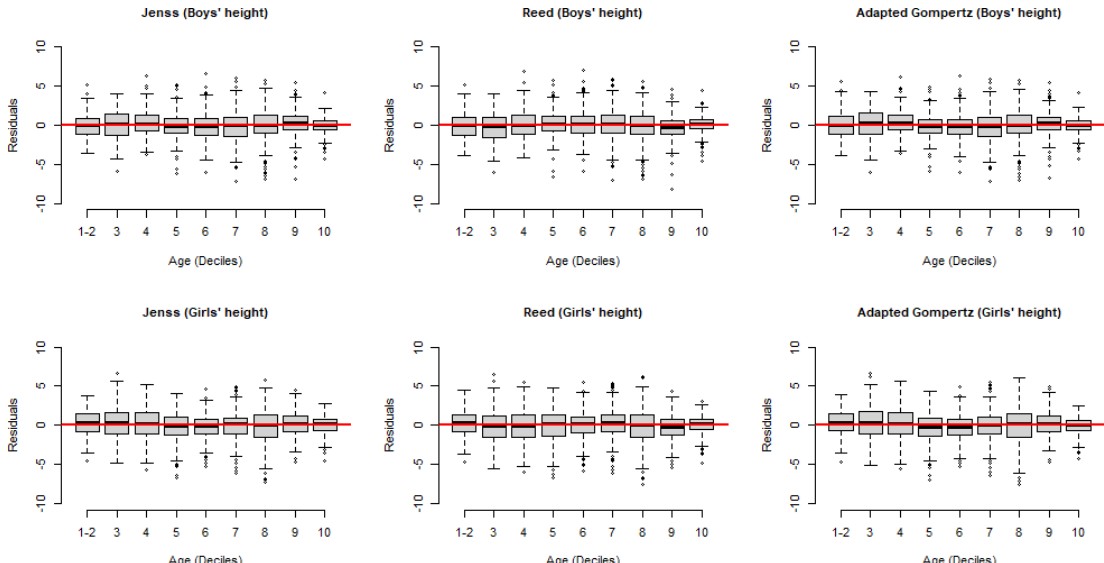

**Figure 2** Residuals of height models for boys and girls (data of the first two deciles was gathered, as there were more than 10% of the data at birth).

For weight models, the Jenss-Bayley model had the lowest AIC and BIC values, both in boys and girls, indicating a better fit than the two other candidate models (table 2). Both the Reed and the adapted Gompertz model had high delta AIC values for both boys (136 and 117, respectively) and girls (129 and 135, respectively), indicating essentially no support for these two candidate model to be better than the best-fitting model (Jenss-Bayley).

For height models, the Jenss-Bayley model and the adapted Gompertz model had the lowest AIC and BIC values for the height model in boys while the Jenss-Bayley model had lower AIC values in girls than the Reed and adapted Gompertz model. Concerning delta AIC values, both the Reed and the adapted Gompertz model had values <10 for boys (8 and 6, respectively), indicating considerably less support for these two models to be the best-fitting model. While in girls, these two candidate models had delta AIC values >10 (28 and

32, respectively), which does not indicate any support for these two models to have a better fit than the Jenss-Bayley model.

Using the Jenss-Bayley model, weight and height were calculated for all the children at age points, particularly when they were not measured, as well as instantaneous growth velocities at different age points, and growth trajectories were calculated (tables 3 and 4, and figure 3, respectively).

Means and SD of parameters estimates of the Jenss-Bayley model along with two other candidate models (the adapted Gompertz model and the Reed model) fitted to weight and height of boys and girls are reported in online supplementary table 3.

Nutritional status for all children was estimated at age 6 years. About one-third of the children were wasted and underweight (online supplementary table 4). While 8% of boys and 4% of girls were stunted (online supplementary table 4). In comparison, in the WHO growth reference

**Table 2** Comparison of the goodness-of-fit of the three candidate models

| Model | Boys | | | | | Girls | | | | |
|---|---|---|---|---|---|---|---|---|---|---|
| | RSD | AIC | BIC | Delta AIC | Log-likelihood | RSD | AIC | BIC | Delta AIC | Log-likelihood |
| Weight | | | | | | | | | | |
| Jenss-Bayley | 0.46 | 5251 | 5300 | | −2613 | 0.46 | 5452 | 5503 | | −2714 |
| Reed | 0.52 | 5387 | 5424 | 136 | −2684 | 0.51 | 5581 | 5619 | 129 | −2781 |
| Adapted Gompertz | 0.49 | 5368 | 5405 | 117 | −2675 | 0.49 | 5587 | 5625 | 135 | −2784 |
| Length/height | | | | | | | | | | |
| Jenss-Bayley | 1.90 | 11786 | 11823 | | −5884 | 2.00 | 12711 | 12749 | | −6347 |
| Reed | 2.00 | 11794 | 11831 | 08 | −5888 | 2.10 | 12739 | 12777 | 28 | −6360 |
| Adapted Gompertz | 1.95 | 11792 | 11829 | 06 | −5887 | 2.10 | 12743 | 12781 | 32 | −6362 |

AIC, Akaike information criterion; BIC, Bayesian information criterion; RSD, residual SD.

**Table 3** Estimated weight and weight growth velocity (SD) of girls and boys aged 0 to 6 years, from the Jenss-Bayley model

| Age | Boys (n=461) | | Girls (n=500) | |
|---|---|---|---|---|
| | Weight, kg | Weight velocity, kg/month | Weight, kg | Weight velocity, kg/month |
| 3 months | 5.6 (0.6) | 0.64 (0.09) | 5.2 (0.6) | 0.57 (0.58) |
| 6 months | 7.1 (0.8) | 0.40 (0.05) | 6.6 (0.7) | 0.38 (0.05) |
| 12 months | 8.8 (0.8) | 0.30 (0.03) | 8.3 (0.8) | 0.21 (0.03) |
| 2 years | 10.6 (0.9) | 0.14 (0.02) | 10.2 (0.9) | 0.12 (0.02) |
| 4 years | 13.7 (1.2) | 0.12 (0.02) | 13.3 (1.2) | 0.12 (0.02) |
| 6 years | 16.6 (1.6) | 0.12 (0.02) | 16.2 (1.7) | 0.12 (0.02) |

population, the percentage of children under −2 z-scores is expected to be 2.3%.

## DISCUSSION

Three growth models (the Jenss, the Reed and an adaptation of the Gompertz model) were fitted to the weight and height of children from birth to age 6 years and their performance was compared. There was no strong trend in the residual distribution of the models over age, suggesting that all the models performed well without systematic underestimation or overestimation of growth at any age. Comparison of AIC/BIC values and RSD supported the conclusion that the Jenss-Bayley model had the best fit. Also, the difference between the AIC (delta AIC) of the Jenss-Bayley model (best model) and other two candidate models demonstrated quantifiable evidence that the Jenss-Bayley model was the best one fitting on weight and height both for girls and boys.[35] Therefore, the Jenss-Bayley model was chosen to estimate mean growth trajectories and instantaneous growth velocities of children, and estimated their nutritional status at age 6 years.

This paper extends the previous limited studies that either fitted the Jenss-Bayley model or compared its goodness-of-fit with other growth models in childhood in SSA. Pagazey and Hauspie[16] successfully fitted the Jenss-Bayley model on Congolese babies weight data. However, the phase of growth studied in this study was restricted from around birth to age 2 years. Similarly, few studies reported Reed model fitted well. Simondon and colleagues[8] compared several models on the growth data of 95 Congolese children and reported that the Reed model had a better fit, but this study did not include the Jenss-Bayley and the adapted Gompertz models and studied child growth only during infancy. Chirwa and colleagues[14] also reported that the Reed model fitted well on the growth data from birth to age 10 years in 453 children living in an urban setting in South Africa. The study reported that the Jenss-Bayley model did not fit well in the early years. However, this study included children only from an urban setting and did not test the performance of the Gompertz model. In summary, the above previous studies from SSA[8 14 16] either tested growth models during a limited phase of growth in early childhood or studied children only from an urban setting. The current study differs from the aforementioned studies in some respects. First, it included growth data of both infancy and early childhood. Second, it included children from a semirural setting which have not been studied before very well in SSA. Third, the growth data used in this study were taken from a prospective study which had standardised measurements of weight and height; therefore, it is likely to be more accurate than data from routine surveys or health records. A limitation of this study is lesser measurements between age 3 and 5 years, but as during this period weight and height are known to increase linearly with age, the impact on the obtained growth modelling parameters is likely to be less. Another limitation of this study is the generalisability of the results. These children lived in a semirural setting with a high prevalence of undernutrition. They also presented a high prevalence of potential risk factors for altered growth[36] (online supplementary table 2). It is therefore possible that their growth would differ from a setting with different characteristics.

The estimated instantaneous growth velocities reflected the overall expected pattern of growth velocity in children that is, a maximum during the earliest age then a decrease until age 2 years and a constant velocity afterwards, as also reported in other population of children,[10 37] although

**Table 4** Estimated length/height and length/height growth velocity (SD) of girls and boys aged 0 to 6 years, from the Jenss-Bayley model

| Age | Boys (n=461) | | Girls (n=500) | |
|---|---|---|---|---|
| | Length/height, cm | Length/height velocity, cm/month | Length/height, cm | Length/height velocity, cm/month |
| 3 months | 59.9 (1.4) | 2.67 (0.10) | 58.6 (1.4) | 2.55 (0.16) |
| 6 months | 66.3 (2.8) | 1.66 (0.06) | 64.8 (3.0) | 1.66 (0.08) |
| 12 months | 73.4 (2.9) | 0.91 (0.06) | 72.1 (1.9) | 0.92 (0.06) |
| 2 years | 82.6 (3.3) | 0.63 (0.05) | 80.9 (2.1) | 0.65 (0.04) |
| 4 years | 96.6 (3.0) | 0.61 (0.05) | 95.9 (2.9) | 0.62 (0.04) |
| 6 years | 111.24 (4.1) | 0.61 (0.05) | 110.7 (3.7) | 0.62 (0.04) |

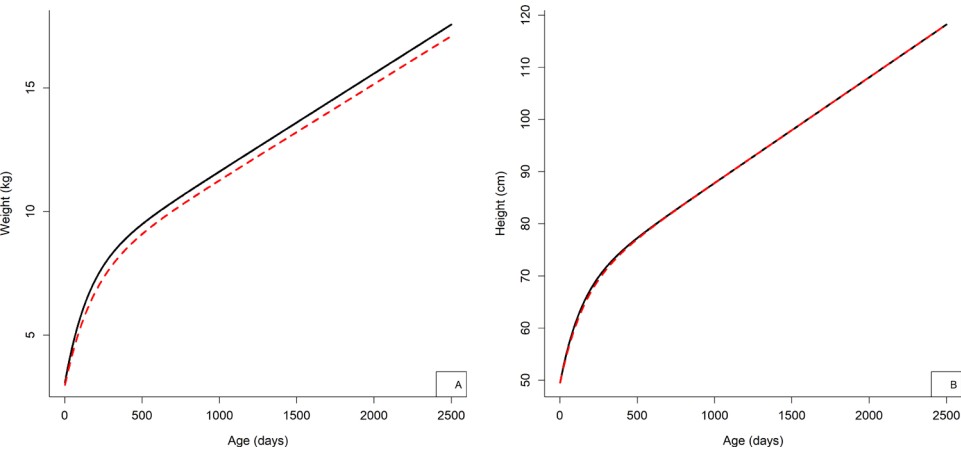

**Figure 3** Weight and height mean growth trajectories from the Jenss-Bayley model. (A) Weight growth curve. (B) Height growth curve. The solid (black) line implies boys' curve and the dashed (red) line implies girls' curve.

growth rate could differ depending on the population, for example, weight velocity among boys at age 3 years (mean±SD): 0.19±0.06 kg/month in American children[37] versus 0.16±0.04 kg/month in French children[10] and weight velocity among boys at 4 years: 0.17±0.02 kg/month in South African children.[38] However, the study by Regnault *et al* excluded birth weight from their analyses while the birth weight was included in the current study, which could have underestimated weight growth velocities during infancy in our population.[28] The mean height growth trajectories were relatively parallel, with no visible difference in the direction of the curve between boys and girls. On the contrary, visible difference in direction was observable in the mean weight growth trajectory of boys and girls. This difference was visible after infancy where the trajectory of girls falls below the curve of boys.

In general, the three models (Jenss-Bayley, adapted Gompertz and the Reed model) seemed to fit well both on weight and height data as was evidenced by mean residuals close to zero (figures 1 and 2). Certain factors could influence the models' goodness-of-fit when comparing several models. For example, a temporary delay to reach the maximum growth velocity after birth might cause difficulty in modelling weight by models like the Jenss-Bayley that do not consider this pattern. The adapted Gompertz model was tested as a potential way to tackle this issue but our adaptation did not perform better than the Jenss-Bayley model. Further analysis was done to show whether predictions by the three models (Jenss-Bayley, Reed and adapted Gompertz model) at different age points that is, at 3 months, 1 year, 3 years and 6 years were different. The predictions differ mainly at 3 months. with no major differences afterwards and the goodness-of-fit indicators supported the use of Jenss-Bayley model (data not shown).

There were slight variations in the way the models converged. The Jenss-Bayley model converged easily on weight data using an unstructured variance-covariance matrix for random effects. More convergence issues were faced after attempting to converge the Jenss-Bayley

model on the height data when the number of iterations was extended to 20 000. The Reed model and the adapted Gompertz model also had convergence issues both on weight and height data with an unstructured variance-covariance matrix but were able to converge with a diagonal variance-covariance matrix for random effects. Although an unstructured variance-covariance matrix would have been preferred for all the models, there was no difference in the fit indicators (ie, AIC/BIC and RSD), fixed effects and predictions (data not shown) when using a diagonal variance-covariance matrix on the Jenss-Bayley model. In particular, a diagonal variance-covariance matrix for random effects does not constraint the growth parameters to be uncorrelated, as the fixed part (population parameters) are free to be correlated anyway.

Convergence issues could be affected by several reasons including, but not limited to, the number of measurement occasions and how the time intervals between measurement occasions are spaced, complexity of the model (eg, the number of parameters and monotonicity), parameterisation (eg, addition of higher-order terms such as ln (age)),[14] as well as the type of statistical packages. Simpler methods than SAEMIX could have been used for the Reed model, but for consistency in methods, it was preferred to fit them with the same one. Some computational and convergence issues we faced could be explained by the limited measurement occasions and unequally spaced time intervals between age 2 and 6 years as also reported by a previous study in South Africa.[14] To facilitate the convergence of the models, constraints of positivity on the parameters of all three models were applied by using exponential functions, as shown in equation 2 for the Jenss-Bayley model.[28]

## CONCLUSION

This paper demonstrated that the Jenss-Bayley model presented the best fit among the three candidate models applied to longitudinal weight and height data of

Beninese children between birth and age 6 years. This model was then used to estimate growth trajectories and growth velocities even for children with few measurements. Finally, it was estimated that about one-third of the children were wasted and underweight at age 6 years. Parameters of this model will be used to study the determinants of growth trajectories in children.

**Author affiliations**
[1]Université de Paris, Centre of Research in Epidemiology and Statistics /CRESS, INSERM, INRA, Paris, France
[2]EHESP, F-35000 Rennes, France
[3]Institut de Recherche pour le Développement (IRD), Cotonou, Benin
[4]MERIT (Mère et Enfant Face aux Infections Tropicales)-UMR 216, Institut de Recherche pour le Développement (IRD), Université Paris Descartes, Paris, France
[5]Paediatric Department, Mother and Child University and Hospital Center (CHU-MEL), Cotonou, Benin
[6]Faculté des Sciences de la Santé, Université d'Abomey-Calavi, Cotonou, Littoral, Benin
[7]EPI-PHARE Scientific Interest Group in Epidemiology of Health Products, French National Agency for the Safety of Medicines and Health Products and the French National Health Insurance, Saint-Denis, Ile-de-France, France

**Acknowledgements** The authors gratefully acknowledge families enrolled in the above-mentioned studies and the contributors of the research team and staff.

**Contributors** SA, JB, and FBL collaborated in the conduct, analysis, interpretation of data and writing. SA performed growth modelling under the supervision of JB. FBL, RZ, AG, DC, MC, AM, JA and NF contributed towards data collection and acquisition; follow-up of studies; revision of the paper; and providing technical support. FBL and JB collaborated in the conception, planning, methodology, overall supervision and critical revision of the work. All authors read and approved the final manuscript.

**Funding** The Malaria in Pregnancy Preventive Alternative Drugs (MiPPAD) study was co-funded by the European and Developing Countries Clinical Trials Partnership (EDCTP; IP.2007.31080.002), the Malaria in Pregnancy Consortium and the following national agencies: Instituto de Salud Carlos III (PI08/0564), Spain; Federal Ministry of Education and Research (BMBF FKZ: da01KA0803), Germany; Institut de Recherche pour le Developpement (IRD), France. The 'Anaemia in Pregnancy: Etiologies and Consequences' (APEC) study received funding from a grant from the Bill & Melinda Gates Foundation. The TOLIMMUNPAL project was funded by Agence Nationale de la Recherche (ANR). The Eunice Kennedy Shriver National Institute of Child Health & Human Development (NIH/NICHD) funded The TOVI study (grant no. R21-HD060524). The EXPLORE study was funded by Fondation de France. Shukrullah Ahmadi received funding from a grant from Fondation de France (reference 00079816).

**Competing interests** None declared.

**Patient consent for publication** Not required.

**Ethics approval** The MiPPAD trial was approved by the institutional review board of the University of Abomey-Calavi, Benin. The APEC study was approved by the Ethics Committee of the Faculty of Medicine of Cotonou, Benin. The TOLIMMUNPAL study was approved by the Comité Consultatif de Déontologie et d'Éthique (CCDE) of the Institut de Recherche pour le Développement (IRD, France) and the Ethics Committee of the Faculté des Sciences de la Santé, Université d'Abomey Calavi, Benin (approval number: 43/11/2010/CE/FSS/UAC). The EXPLORE study was approved by the Research Ethics Committee of Institute of Applied Biomedical Sciences, Benin (Comité d'Ethique de la Recherche, Institut des Sciences Biomédicales Appliquées, CER-ISBA, Benin) (approval number: 87 du 17/05/2016). The TOVI study was approved by the institutional review boards of the University of Abomey-Calavi in Benin and New York University in the United States (IRB#09-1253)

**Provenance and peer review** Not commissioned; externally peer reviewed.

**Data availability statement** Data are available upon reasonable request.

**ORCID iDs**
Shukrullah Ahmadi http://orcid.org/0000-0001-7071-8761
Jérémie Botton http://orcid.org/0000-0002-4814-6370

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
