## [Reviewer comments · BMJ Open]

ARTICLE DETAILS

TITLE (PROVISIONAL)	Comparison of Growth Models to Describe Growth from Birth to 6 Years in a Beninese Cohort of Children with Repeated Measurements
AUTHORS	AHMADI, Shukrullah; Bodeau-Livinec, Florence; Zoumenou, Roméo; Garcia, Andre; Courtin, David; Alao, Jules; Fievet, Nadine; Cot, Michel; Massougbojji, Achille; Botton, Jérémie

VERSION 1 – REVIEW

REVIEWER	Sara Benjamin-Neelon Johns Hopkins Bloomberg School of Public Health, Baltimore, Maryland, USA.
REVIEW RETURNED	06-Jan-2020

GENERAL COMMENTS	Thank you for the opportunity to review this manuscript. This is an interesting paper summarizing results from a study evaluating three distinct growth models: Jenss-Bayley, Reed, and an adaptation of the Gompertz model. The authors use data from multiple cohorts of Beninese children followed from birth to (around) 6 years of age. There are some strengths to the study, including the fact that the authors compared commonly used (or adapted) growth models in longitudinal samples of children. There are also some weaknesses, including limited generalizability of the results and some questions related to the modeling. Please see specific comments below. Comments: 1. It would be helpful to know how many children had measurements beyond 12 months? Beyond 24 months? Also, how many children had measurements at 3, 4, 5, or 6 years? Table 1 is helpful but it does not provide this information.2. It may be safe to assume the random effects are normally distributed, although this is not stated. The authors chose a diagonal covariance, implying independence. Did the authors consider a model that allowed for a subset of the random effects to be correlated (e.g., A_i and B_i in the Reed model)?3. The third full paragraph of page 8 is unclear. What parameters did the authors use to represent average trajectories? Are the velocities obtained directly from the SAEMIX package? It would be helpful to have a bit more detail on this part of the modeling.4. The authors might consider using the small sample AIC, which may be more appropriate for this sample size. Please see work by Burnham and Anderson for more information.5. What final recommendations do the authors make? Should one use the Jenss-Bayley model as the default? How generalizable are
---

	the results, given the caveats about generalizability noted in the Discussion section? 6. To address the comment above, a simulation would be helpful, in which data are generated under the three growth models. Next, for each dataset, fit the three models, for a total of 9 AICs. If the Jenss model performed well even when the data were generated under Reed or Gompertz, this would lend support to its general use. Otherwise, it is difficult to generalize the results beyond this particular application. 7. Page 7, line 51: Should be D_i not D. Also, D should be italicized throughout (minor). 8. Please check inconsistent use of capital letters in title (minor). 9. Some sentences appear to be missing a word or need revision (e.g., lines 22-24 in the Abstract) (minor). 10. The authors switch from first person active voice to passive voice throughout the manuscript, which decreases flow and readability (minor). 11. Page 6, lines 7-9: How many 2-year-olds were measured standing vs. recumbent (minor)?
--	---

REVIEWER	Izzuddin Aris Harvard Medical School and Harvard Pilgrim Health Care Institute
REVIEW RETURNED	03-Feb-2020

GENERAL COMMENTS	Growth models have often been used to summarize and capture intrinsic features of weight and height growth at the individual and population level. Few, however, have substantively assessed the merits of different models in characterizing growth trajectories among African children. The paper by Ahmadi et al compares the performance of three known growth models (i.e. Jenss-Bayley, Reed and an adapted Gompertz model) in fitting repeated weight and length/height measurements between birth and six years from an African birth cohort. Overall, the study is exceedingly well-conducted and I only have the following minor comments: 1. The authors should provide a more holistic overview in the Introduction on the importance of characterizing growth trajectories (e.g., characterizing childhood trajectories of weight, length/height or BMI may be important for surveillance, allow for more accurate identification of modifiable risk factors and prediction of health outcomes. Characterization of these trajectories may also help to identify “sensitive” age periods during which interventions may be especially useful.) 2. While the mean of the residuals are close to zero, there appears to be some heteroscedasticity in the distribution of the residuals. This is evident in models estimating weight, where the residuals appear to have a larger variance at later ages (i.e., deciles 8 and 9) than at earlier ages, which suggest that these models fit “better” at earlier ages than later ages. Could the authors account for this heteroscedasticity in their models, perhaps by estimating residual error parameters over deciles of age? 3. The authors should clearly specify that their study has received
---

	appropriate participant consent and ethics approval.
REVIEWER	Nathalie Costet Irset, Inserm UMR_S 1085 France
REVIEW RETURNED	13-Feb-2020
GENERAL COMMENTS	This paper reports the analysis of growth data of Beninese children aged 0 to 6 years with relevant statistical methods that are correctly used. Fit indicators and check of the statistical assumptions of the growth models show that they describe correctly growth trajectories of this population. However, the manuscript presents some limitations of different nature that should be addressed before being published as a separate paper. As it is, the manuscript appears more as the first step of a future main study about the determinants of growth trajectories and should be completed to produce more generalisable results. ABSTRACT Objective: The objective of the paper is to compare several structural statistical models to describe the growth trajectory of children (0-6 years) in a semi-rural area in Sub-Saharan Africa. The paper does not really address its “usefulness”. I would delete the end of the last sentence. Results: Delete citation of AIC numeric results, as they are useless for the reader (AIC are data specific). The reader only wants to know which model fit the best. You should also mention that undernutrition is estimated at 6 years. Conclusion: Delete the first sentence (already said in Results). INTRODUCTION The way the existing literature is briefly presented (Page 4, Lines 35-42) is very confusing and has to be rewritten to be understood. It mixes citations of studies conducted in low- and middle-income countries (not only African countries), considerations about the type of data (longitudinal or not) and types of models used (in African studies, or globally ?). It is said that “very few have collected data longitudinally”, followed by one unique reference 28. But reference 7 also includes longitudinal data (as an example). The term “in addition” is confusing (other studies ? some of them are already cited) It is also said that “the comparative merits of these models have not been assessed”, which is not true (see reference 21). Moreover, references 8 and 21 are the same ! METHODOLOGY Study design and data sources The way the final sample is derived is not well documented. The present study includes children from different successive studies. Five names of studies are cited, some of them are linked to a reference, but the TOVI study is not referenced, for example. The characteristics of the children are not mentioned, when some characteristics may highly impact growth trajectories (birth weight, prematurity, presence of pathologies ...). The selection criteria along the follow-up is not described. A flow chart showing how the different studies are embedded has to be presented, at least in supplemental files. Data inclusion Children with at least 2 measurements were included. Differential numbers of measurements are not problematic for mixed growth

	models, but it supposes that children participating in each follow-up are comparable (randomly selected). As the selection process of the sample is not clearly presented and the description of the characteristics of the children is missing, it is difficult to evaluate whether the hypothesis is reasonable. Raw growth curves have been checked visually. Did you make some corrections (how much) ? The choice of the 3 models tested is not motivated. Why selecting these three ones among numerous other? Why correcting the Gompertz model? (it is explained in the discussion, but too late). The citation of 2 papers (Tjorve 2017, Winsor 1932) suggests that the correction has been proposed previously, but the authors say (line 36) "WE expanded the Gompertz..."). So are you the first to use this correction ? Page 7, line 3: ASYMPTOTIC and not asymptomatic slope. Equations 1, 2: There are some typographic problems with superscripts (shifts) Equation 3 : t_{ij} and not t_j Statistical methods Page 8 , lines 10-12 : lm and nls functions are R functions (it should be specified) Was a diagonal variance-covariance matrix selected for random effects for the 3 models ? This constraints the growth parameters of the models to be uncorrelated, which is not common, and not so realistic, as phases of growth are usually correlated. This is something that may be commented on in the discussion section. The paragraph describing the estimation of the mean growth trajectories (lines 22-39 page 8) is confusing. It is not clear how the mean growth trajectory is calculated : "A set of parameters obtained from the mixed-effects model were used to represent average growth trajectories". Which set of parameters ? The following sentence can not be understood without reading the cited Botton's paper. This paragraph should be understood without access to this paper. Please clarify. Line 42 page 8 : undernutrition WAS estimated RESULTS The first sentence of this paragraph should appear in the Methods section. Again, I am surprised that no descriptive data of the population under study are presented. Growth data are considered without any health/clinical or socio-economic context, which limits the use, comparison of these results for other studies and the generalizability of the results. I think that minimal information should be presented. Page 9, line 37 : Figure 1 and figure ??? Page 10, line 26-28: where THEY WERE not measured Table 3, title : ESTIMATED WEIGHT AND weight growth velocity(Jenns-Bayley model) Table 3, columns title should be : "Weight" and "Weight velocity" ("Estimated" is in the title and both are estimated) Same remarks in Table 4. The commentary Page 11, lines 21-22 is confusing. I understand that as the variance of the parameters is low compared to their estimation, the mean trajectory is reliable, but I don't understand why it suggests that "all the parameters highly contributed to the mean trajectory". Parameters of the growth curves correspond to specific phases of growth and if some specific parameters had larger variances, one could comment on the phases of the mean growth that are less reliable than others. The sentence is too allusive. Authors should mention which conclusions they draw from that and
--	--

	why. And this should be placed in the Discussion section and not in the Results. Comparison of models was made only on RSD, AIC, BIC and log-likelihood statistics. These are the relevant indicators to use. But they are not very concrete for readers. It may interesting to produce differences observed at some particular ages (3 months, 1 year, 3 years, 6 years, for example), in order to measure the size of the differences between the different models. The reader may also be interested in the comparison of performance between the original Gompertz model and the corrected version. DISCUSSION Page 11, line 55 : nutritional status AT 6 YEARS. I think that discussion of the existing literature should focus on studies conducted in Sub-Saharan Africa, in populations potentially affected by undernutrition, as this is the main value of this present study: how popular growth models - mainly used and developed in developed countries - can fit growth in specific populations with a high prevalence of impaired growth? Page 12, lines 40-42: Differences in growth and nutritional status of children living in urban or rural settings are suggested. In which direction? Page 12, Line 45: A strength of the study is that standardized measurements of weight and height were available (not only their prospective collection) Page 12, Line 47: Regarding the number of measurements, see my previous remark in the Method section. Page 13, lines 6-10: I don't understand what the authors mean when saying "On the other hand, reduced measurement errors ... may decrease residuals ..." ! Please clarify. To improve the generalizability of the results of this study, models may be further compared regarding the interpretability of their parameters, and computational, parameterization and convergence issues. The models differ by the number of their parameters and the specific phases of growth that can describe. Depending on the future use of such models to study growth determinants, some of them may be more relevant, at comparable level of fit. CONCLUSION Last sentence : PARAMETERS of this model
--	---

VERSION 1 – AUTHOR RESPONSE

Reviewer: 1

The first reviewer wrote:

Thank you for the opportunity to review this manuscript. This is an interesting paper summarizing results from a study evaluating three distinct growth models: Jenss-Bayley, Reed, and an adaptation of the Gompertz model. The authors use data from multiple cohorts of Beninese children followed from birth to (around) 6 years of age. There are some strengths to the study, including the fact that the authors compared commonly used (or adapted) growth models in longitudinal samples of children.

Reply: Thank you for the praise

Comment: 1. It would be helpful to know how many children had measurements beyond 12 months?

Beyond 24 months? Also, how many children had measurements at 3, 4, 5, or 6 years? Table 1 is helpful but it does not provide this information.

Reply: Thank you for raising this point. We included the following information in a supplementary table in the manuscript. We also added a flow chart in supplemental information to help the understanding of the measurements because of the different ancillary studies in this cohort:

“Supplementary Figure 1 shows participants inclusion and how different ancillary studies are embedded in this cohort.”

Supplementary Table 1. Number of children with anthropometric measurements at different age intervals. Data from boys (n=461) and girls (n=500) from birth to around six years of age

We also included in the DATA INCLUSION section of the manuscript:

“The number of children with anthropometric measurements at different age intervals are presented in the supplementary Table 1”

There were few measurements between 3 and 5 years, but this is not a major issue to estimate growth trajectories because this is a period of linear growth.

Comment: 2. It may be safe to assume the random effects are normally distributed, although this is not stated. The authors chose a diagonal covariance, implying independence. Did the authors consider a model that allowed for a subset of the random effects to be correlated (e.g., A_i and B_i in the Reed model)?

Reply: Indeed, model assumptions include that individual parameters are independent and normally distributed, residual errors are independent and normally distributed. The residual variance is constant. We've checked these assumptions graphically. We did consider an unstructured covariance matrix and a subset of random effects to be correlated but due to convergence issues, we finally used a diagonal matrix. The effect of using different types of covariance structures does not strongly affect the individual trajectories, but rather the variances. As we are interested in the individual predictions, not by statistical inference, this does not constitute an important limitation.

Comment: 3. The third full paragraph of page 8 is unclear. What parameters did the authors use to represent average trajectories? Are the velocities obtained directly from the SAEMIX package? It would be helpful to have a bit more detail on this part of the modeling.

Reply: Firstly, fixed-effect parameters (A, B, C and D) obtained from the Jenss-Bayley model using SAEMIX package were used to represent mean trajectories. Secondly, individual velocities were obtained using the following equation which is now added in the manuscript. So, the velocities were not obtained directly from the SAEMIX package but the model parameters obtained from the SAEMIX package were substituted for each child in the following equation. This is now added in the manuscript:

“As a derivative of the Jenss-Bayley model, the following equation can be used to estimate individual growth velocity against time [1].

$$dy/dt = \exp(-B_i + \exp(C_i - D_i - \exp(-D_i \cdot t_{ij}))$$

Comment: 4. The authors might consider using the small sample AIC, which may be more appropriate for this sample size. Please see work by Burnham and Anderson for more information.

Reply: We used both AIC and BIC to select the models and there were no discrepancies. Our conclusions are still admissible even if the sample size is not too large because BIC values are not biased even if the sample size is small. Additionally, the BIC defined in the SAEMIX package [2] uses a formula to estimate corrected BIC (BICc) to better account for the mixed-effect models specificities.

Furthermore, our sample size was large enough with a ratio of $n/k > 40$, where "n" denotes the sample size and "k" denotes the number of parameters. For example n/k for the Jenss-Bayley Height model for boys = $2514/9 = 279.3$. In that case, AICc and AIC will give similar values (see below), hence the calculation of corrected AIC is not particularly interesting here and there will be no disadvantage in using AIC as recommended by Burnham and Anderson [3].

This is shown in the following example for the calculation of corrected AIC (AICc) value for the AIC of the Jenss-Bayley Height model for boys :

The corrected AIC (AICc) proposed by Hurvich and Tsai (1989) and referenced by Burnham and Anderson, includes a correction for small sample sizes as follows:

$$AICc = AIC + 2k(k+1)/(n-k-1)$$

where "n" denotes the sample size and "k" denotes the number of parameters.

$$AICc = 11786 + 2 \cdot 9(9+1)/(2514-9-1)$$

$$AICc = 11786$$

Therefore $AIC = AICc$

Comment: 5. What final recommendations do the authors make? Should one use the Jenss-Bayley model as the default? How generalizable are the results, given the caveats about generalizability noted in the Discussion section?

Reply: Indeed, Jenss-Bayley model should be used in this population of children based on the goodness-of-fit indicators. However, in different populations, results could be different. So our advice is to check the goodness-of-fit of different models in different populations and choose the model with the best goodness-of-fit.

Comment: 6. To address the comment above, a simulation would be helpful, in which data are generated under the three growth models. Next, for each dataset, fit the three models, for a total of 9 AICs. If the Jenss model performed well even when the data were generated under Reed or Gompertz, this would lend support to its general use. Otherwise, it is difficult to generalize the results beyond this particular application.

Reply: The sampling was not representative of the Beninese population. Our sample comes from an HIV-negative population in a rural setting where many health problems (e.g. undernutrition, infectious diseases) are common. Therefore, we acknowledge that the generalizability of our results is limited. Other studies using subjects from settings other than this region (for example from urban setting) will be useful to compare the consistency of the results. The idea behind this study was not generalizability but rather finding the best model among potentially good models in this particular population of children with a high prevalence of malnutrition and to study a new model (adapted Gompertz model). This is why we did not perform simulations. As requested by reviewer 3, we added some characteristics of our population (Supplementary Table 2). This may be useful in the future for other studies to compare their populations with ours.

We also added in the discussion:

“Another limitation of this study is the generalizability of the results. These children lived in a semi-rural setting with a high prevalence of undernutrition. They also presented a high prevalence of potential risk factors for altered growth [4] (Supplementary Table 2). It is therefore possible that their growth would differ from a setting with different characteristics.”

Comment: 7. Page 7, line 51: Should be D_i not D. Also, D should be italicized throughout (minor).

Reply: Corrections are made in the manuscript:

“...parameter B_i is the asymptotic slope or growth rate/velocity of preschool growth, C_i is an important component of rapid early childhood growth[5, 6] and models the exponential growth deceleration, D_i is used to better model early life variations in growth, and A_i is the intercept”

Comment: 8. Please check inconsistent use of capital letters in title (minor).

Reply: Corrections are made in the manuscript:

“Comparison of Growth Models to Describe Growth from Birth to 6 Years in a Beninese Cohort of Children with Repeated Measurements”

Comment: 9. Some sentences appear to be missing a word or need revision (e.g., lines 22-24 in the Abstract) (minor).

Reply: Corrections are made in the manuscript:

“The goodness-of-fit of the models was assessed using residual distribution over age and compared with the Akaike Information Criterion (AIC) and Bayesian Information Criterion (BIC). The best-fitting model allowed estimating mean weight and height growth trajectories, individual growth and growth velocities”

Comment: 10. The authors switch from first person active voice to passive voice throughout the manuscript, which decreases flow and readability (minor).

Reply: Passive voice is now used throughout the manuscript

Comment: 11. Page 6, lines 7-9: How many 2-year-olds were measured standing vs. recumbent (minor)?

Reply: All of the children were measured standing from 2 years onwards. Amendments are now made in the manuscript:

“From 2 years onwards, standing weight and height was measured for all children.”

Reviewer: 2

The second reviewer wrote:

Growth models have often been used to summarize and capture intrinsic features of weight and height growth at the individual and population level. Few, however, have substantively assessed the merits of different models in characterizing growth trajectories among African children. The paper by Ahmadi et al compares the performance of three known growth models (i.e. Jenss-Bayley, Reed and an adapted Gompertz model) in fitting repeated weight and length/height measurements between birth and six years from an African birth cohort. Overall, the study is exceedingly well-conducted and I only have the following minor comments:

Reply: Thank you for the praise

Comment: 1. The authors should provide a more holistic overview in the Introduction on the importance of characterizing growth trajectories (e.g., characterizing childhood trajectories of weight, length/height or BMI may be important for surveillance, allow for more accurate identification of modifiable risk factors and prediction of health outcomes. Characterization of these trajectories may also help to identify “sensitive” age periods during which interventions may be especially useful.)

Reply: We expanded the introduction by providing a more holistic overview of growth trajectories in the introduction:

“Mathematical growth modelling is a powerful tool for the study of child growth and growth trajectories. It consists of fitting models to physical growth data (e.g. weight, length/height, head circumference) with the aim of obtaining an appropriate growth curve that will conveniently summarize growth information provided by weight and height measurements of children, even from irregularly spaced

growth measurements [5]. Clinicians routinely compare growth data to growth charts to identify impaired growth trajectories in children. Indeed, looking at growth trajectories (weight, height/length or BMI) has gained importance for several purposes. It is becoming essential for surveillance to identify abnormal growth trajectories[7]. In the post-natal period, length/height and weight surveillance is an essential tool for monitoring child growth. It is also used in research for the purpose of studying determinants of growth including identification of modifiable risk factors and sensitive age periods during which interventions may be especially useful to achieve optimal growth [8]. Characterizing growth trajectories is also important to study whether past growth is associated with future growth patterns or later health outcomes [9]"

We also developed further the introduction by elaborating on growth models:

"Two of the three models are amongst the most common structural models described in the literature, while non-structural models are mainly polynomial and splines [10-13] as cited in Chirwa et al. (2014) [14]. The parameters of structural models have biological interpretations. In contrast, non-structural models which may fit better overall tend to demonstrate instability at extremities and do not formulate any particular form of growth curve. In addition, the parameters obtained from non-structural models do not provide any biological interpretation [12, 15]"

Comment: 2. While the mean of the residuals are close to zero, there appears to be some heteroscedasticity in the distribution of the residuals. This is evident in models estimating weight, where the residuals appear to have a larger variance at later ages (i.e., deciles 8 and 9) than at earlier ages, which suggest that these models fit "better" at earlier ages than later ages. Could the authors account for this heteroscedasticity in their models, perhaps by estimating residual error parameters over deciles of age?

Reply: As the reviewer suggested about larger variances of the residuals at later ages in the weight models, we estimated standard deviations (SD) over deciles of age for all weight models and present them in the following table.

Table 1. SD (n) over deciles of age for weight models

So if there is an increase in the SD, this is not a constant increase and therefore a model on the variance may not take this into account very well.

Despite of that, we also attempted to handle this by re-converging models using proportional residual error model in SAEMIX package in R but it did not improve the heteroscedasticity and the predictions did not change substantially.

Comment: 3. The authors should clearly specify that their study has received appropriate participant consent and ethics approval.

Reply: Corrections are made in the study design and data sources:

"Appropriate participant consent and ethics approval were obtained for the different studies included."

Reviewer: 3

The third reviewer wrote:

"This paper reports the analysis of growth data of Beninese children aged 0 to 6 years with relevant statistical methods that are correctly used. Fit indicators and check of the statistical assumptions of the growth models show that they describe correctly growth trajectories of this population"

Reply: Thank you for the praise

Comment: "However, the manuscript presents some limitations of different nature that should be addressed before being published as a separate paper. As it is, the manuscript appears more as the

first step of a future main study about the determinants of growth trajectories and should be completed to produce more generalisable results.”

Reply: Thank you for pointing out to the limitations of this study. We fully acknowledge them and revised the manuscript.

ABSTRACT

Comment: Objective: The objective of the paper is to compare several structural statistical models to describe the growth trajectory of children (0-6 years) in a semi-rural area in sub-Saharan Africa. The paper does not really address its “usefulness”. I would delete the end of the last sentence.

Reply: As suggested we removed the line about its “usefulness” but replaced with the characteristics of the growth:

“Objective To select a growth model that best describes individual growth trajectories of children and to present some growth characteristics of this population.
”

Comment: Results: Delete citation of AIC numeric results, as they are useless for the reader (AIC are data specific). The reader only wants to know which model fit the best. You should also mention that undernutrition is estimated at 6 years.

Reply: We removed the AIC numerical values and added in the results:
“Undernutrition was estimated at 6 years of age”

Comment: Conclusion: Delete the first sentence (already said in Results).

Reply: We removed the first sentence from the conclusion and reformulated the conclusion:
“The growth parameters of the best fitting Jenss-Bayley model can be used to describe growth trajectories and study their determinants”.

INTRODUCTION

Comment: The way the existing literature is briefly presented (Page 4, Lines 35-42) is very confusing and has to be rewritten to be understood. It mixes citations of studies conducted in low- and middle-income countries (not only African countries), considerations about the type of data (longitudinal or not) and types of models used (in African studies, or globally?).

Reply: We have re-written the description of existing literature in the introduction:
“Many studies have analyzed child growth data in Africa, but few studies[13, 14, 16-20] have applied quadratic, structural or non-structural growth models on African longitudinal growth data. A summary of some of these studies are presented in Table 1. Furthermore, of these studies only two studies [14, 17] assessed the comparative merits of several models on child growth in SSA to find one that best describes the particular population.”

As highlighted by the reviewer, we decided to cite only references that used longitudinal data and used quadratic, structural or un-structural growth models on African population:

Table 1 (manuscript). Summary of studies that have used quadratic, structural and non-structural models to describe physical growth.

Comment: It is said that “very few have collected data longitudinally”, followed by one unique reference 28. But reference 7 also includes longitudinal data (as an example). The term “in addition” is confusing (other studies ? some of them are already cited)

Reply: We have re-written this part about the longitudinal studies in Africa. All the studies cited [13,

14, 16-20] are longitudinal studies:

“Few studies [13, 14, 16-20] have applied quadratic, structural or non-structural growth models on African longitudinal growth data”

Comment: It is also said that “the comparative merits of these models have not been assessed”, which is not true (see reference 21). Moreover, references 8 and 21 are the same !

Reply: In reference 21, (Simondon et al. 1992) compared several models but their study did not assess the comparative merits of the Jenss-Bayley model and the new adapted Gompertz model.

We rephrased this sentence about “the comparative merits of these models...”

“However, there are limited studies [14, 17] in the literature that have assessed the comparative merits of models like the Jenss-Bayley on child growth in SSA. Furthermore, as to our knowledge, no study has compared the fitness of the adapted Gompertz growth model on child growth data”.

Also, we should specify that the adapted Gompertz, as to our knowledge, has never been compared because we propose this model in the present paper for the first time.

We corrected the duplicate reference (Simondon et al., 1992). Thanks for pointed this mistake out.

METHODOLOGY

Study design and data sources

Comment:

The way the final sample is derived is not well documented. The present study includes children from different successive studies. Five names of studies are cited, some of them are linked to a reference, but the TOVI study is not referenced, for example. The characteristics of the children are not mentioned, when some characteristics may highly impact growth trajectories (birth weight, prematurity, presence of pathologies ...). The selection criteria along the follow-up is not described. A flow chart showing how the different studies are embedded has to be presented, at least in supplemental files.

Reply: We now cited a reference for the TOVI study: Mireku et al. (2015)

We revised and developed further the selection criteria and follow-up for each study in the methodology:

“Participating children were born of HIV-negative mothers (n = 1182) enrolled in the MiPPAD clinical trial (Malaria in Pregnancy Preventive Alternative Drugs) (NCT00811421) in 2011 in the district of Allada, south Benin [21]. The MiPPAD trial, which enrolled pregnant women before 28 weeks of gestation at the first antenatal care visit, compared the efficacy of two intermittent preventive treatments for malaria in pregnancy (IPTp). Height and weight of children were assessed at birth and women were requested to bring their infants to the health center when the babies were 1 month, and also at 9 and 12 months of age. Children included in our analyses met the following inclusion criteria: being born within the MiPPAD clinical trial, and assessed at least twice with valid anthropometric measurements (weight and height) between birth and six year of age within MiPPAD, APEC, TOLIMMUNPAL, TOVI or EXPLORE studies.”

“The APEC study included the first 400 offspring born within the MiPPAD study [34] including anthropometric measurements of children at 6 and 9 months.”

“Infants assessed in the APEC study were also followed-up in the TOLIMMUNPAL study between birth and 24 months with further anthropometric measurements at 15, 18, 21 and 24 months.”

“The TOVI study followed-up at one year-of-age 747 singleton children born within MiPPAD. Among these children, only 92 children randomly selected were assessed between 3 and 5 years of age for anthropometric measurements”

“Finally, at six years of age, children were assessed within the EXPLORE study for anthropometric measurements (2016 -2018) [22, 23].”

For the baseline characteristics of children, we now added the following supplementary table: Supplementary Table 2. Summary of maternal and child characteristics in the prospective child cohort in Benin, sub-Saharan Africa.

We also added the following sentences in the RESULTS section:

“Some descriptive characteristics of the study population are presented in the Supplementary table 2.”
“The overall mean maternal age at 1 year of the study child was ≈ 25.8 years (s.d. 5.6). Almost $\frac{3}{4}$ of mothers could not read or write at the start of cohort. Almost 17% of mothers were underweight before their pregnancy. Almost 10% of boys and 14% of girls had a lower birth weight (< 2.5 kg) and less than 7 % of children were born preterm. Almost 10% of children had malaria at 1 year of age and almost 74% of boys and 67% of girls had anemia at 1 year of age.”

We also proposed a flow chart (Supplementary Figure 1) showing how studies are embedded in each other.

Data inclusion

Comment: Children with at least 2 measurements were included. Differential numbers of measurements are not problematic for mixed growth models, but it supposes that children participating in each follow-up are comparable (randomly selected). As the selection process of the sample is not clearly presented and the description of the characteristics of the children is missing, it is difficult to evaluate whether the hypothesis is reasonable.

Reply: We now described the selection process of the population in the manuscript (response in the above comment).

All children were born within the MiPPAD trial. Sub-populations from APEC, TOLIMMUNPAL and TOVI were randomly selected. The first 400 infants to be delivered from these women, were enrolled in APEC and TOLIMMUNPAL from January 2010 until June 2011 and followed throughout the first year of life [4]. Therefore, the inclusion of these studies should not affect the characteristics of the children. The included flow chart brings clarification on the selection of the population and follow-up.

Comment: Raw growth curves have been checked visually. Did you make some corrections (how much) ?

Reply: We now added in the data inclusion:

“65 height (out of 5291 measures) and 35 weight observations (out of 5291 measures) were excluded. They were considered as outliers on the descriptive individual weight and height plots even after the systematic removal of observations with WAZ or HAZ greater than +4 or less than -4”

Comment: The choice of the 3 models tested is not motivated. Why selecting these three ones among numerous other

Reply: The aim of our growth modelling is to describe at best individual growth trajectories of the children. These three models have been selected for the following reasons:

1. The foremost important rationale behind comparing these three models is because these are the most common structural models that fit well on growth data. Two of the three structural growth models (the Jenss-Bayley and the Reed model) have previously been shown to fit well to the infancy or childhood period [12]. We did not include Count model because the Reed model is an extension of the Count Model.
2. We also choose these structural models because parameters define a particular form of the growth curve from a biological basis. In contrast, non-structural models lack these advantages and may demonstrate instability at extremities
3. Other models could have been tested but the overall fit of the selected models was very optimal and they present advantages for future studies in this population

We also further developed the introduction by elaborating on structural and non-structural growth models, also in response to reviewer 2:

“These three models are amongst the most common structural models described in the literature, while non-structural models are mainly polynomial and splines [10-13], as cited Chirwa et al. (2014) [14]. The parameters of structural models have biological basis. In contrast, non-structural models do not formulate any particular form of growth curve and may demonstrate instability at extremities. In addition, the parameters obtained from non-structural models do not provide any biological interpretation [12, 15]”

Comment: Why correcting the Gompertz model? (it is explained in the discussion, but too late).

Reply: We now added the following in the Methodology under the sections of Growth Models: “This adaptation allows the model to have a non-flat asymptote (slope=velocity different from 0). This permits to fit the linear part of the growth at the later ages in our data (constant linear growth velocity)”

Comment: The citation of 2 papers (Tjorve 2017, Winsor 1932) suggests that the correction has been proposed previously, but the authors say (line 36) “WE expanded the Gompertz...”). So are you the first to use this correction?

Reply: As pointed out by reviewer # 3, previously few re-parameterizations or corrections have been suggested by different authors as mentioned in the paper by Tjorve (2017). But those re-parameterizations have been used typically in biological and organismal growth including microorganisms, cancer cell growth, and in animals. The adaption of the Gompertz model we proposed in this paper has never been published previously, to our knowledge, on human growth. As also explained above, “this allows the model to have an asymptote which is not flat (slope=velocity=0). This permits to fit the linear part of the growth at the later ages in our data (constant linear growth velocity)”

Comment: Page 7, line 3: ASYMPTOTIC and not asymptomatic slope.

Reply: Corrections are now made in the manuscript:

“parameter B_i is the asymptotic slope or growth rate/velocity”

Comment: Equations 1, 2: There are some typographic problems with superscripts (shifts)

Reply: Corrections are now made in the manuscript:

$$E(y_{ij}) = a_i + b_i \cdot t_{ij} - (\exp c_i + d_i \cdot t_{ij}) \quad (1)$$

$$E(y_{ij}) = \exp A_i + \exp -B_i \cdot t_{ij} + \exp C_i \cdot (1 - \exp -D_i \cdot t_{ij}) \quad (2)$$

Comment: Equation 3 : t_{ij} and not t_j

Reply: Corrections are now made in the manuscript:

$$E(y_{ij}) = A_i \cdot \exp (-B_i \cdot \exp (-C_i \cdot t_{ij})) + D_i \cdot t_{ij} \quad (3)$$

Comment:

Statistical methods

Page 8 , lines 10-12 : `lm` and `nls` functions are R functions (it should be specified)

Reply: We now cited R in our manuscript and added the following citation:

R Development Core Team (2017). R: A language and environment for statistical computing. R Foundation for Statistical Computing, Vienna, Austria. URL: <http://www.R-project.org>.

Comment: Was a diagonal variance-covariance matrix selected for random effects for the 3 models ? This constraints the growth parameters of the models to be uncorrelated, which is not common, and not so realistic, as phases of growth are usually correlated. This is something that may be commented

on in the discussion section.

Reply: An unstructured variance-covariance matrix was selected for random effects for the Jenss-Bayley weight model. However, we faced convergence issues after trying an unstructured variance-covariance matrix for the Jenss-Bayley height model, the Reed and the adapted Gompertz model. Therefore, we finally used a diagonal matrix for the latter two models.

It should be noted that using diagonal variance-covariance matrix for random effects does not constraint the growth parameters to be uncorrelated. The four model parameters are free to vary for each individual so using a diagonal variance-covariance matrix should not strongly affect individual trajectories. According to Littell et al. (2000) there is no effect of using different variance-covariance structure on the fixed effects (i.e. growth parameters of the models in the current study) but standard errors [24]. To compare the three models with the same covariance structure, Jenss-Bayley weight models were also fitted using a diagonal matrix and we found no major difference in the model outputs (i.e. AIC/BIC and RSD); and fixed effects (data not shown). Therefore, the Jenss-Bayley model was the best performing model even when the diagonal covariance matrix was used for the three models as well.

So variances could have been impacted but this is not a problem here as we adopted a two-steps growth modelling approach: once the parameters are predicted we use them to calculate predictions of anthropometric parameters or velocities.

Comment: The paragraph describing the estimation of the mean growth trajectories (lines 22-39 page 8) is confusing. It is not clear how the mean growth trajectory is calculated : “A set of parameters obtained from the mixed-effects model were used to represent average growth trajectories”. Which set of parameters? The following sentence can not be understood without reading the cited Botton's paper. This paragraph should be understood without access to this paper. Please clarify.

Reply: As the reviewer suggested we reformulated this paragraph
“Fixed-effect parameters (A, B, C and D) obtained from the mixed-effects model using SAEMIX package were used to represent average growth trajectories. The Jenss-Bayley model then allowed predicting individual weight and height by substituting the individual model parameters into their corresponding model equation (equation 2). Similarly, individual growth velocities were calculated by substituting determined individual parameters into the derivative of the Jenss-Bayley model. As a derivative of the Jenss-Bayley model, following equation can be used to estimate growth velocity over time.

$$dy/dt = \exp-Bi + \exp Ci-Di - \exp -Di \cdot tij$$

Comment: Line 42 page 8: undernutrition WAS estimated

Reply: Corrections are now made in the manuscript:
“undernutrition was estimated”

Comment:
RESULTS

The first sentence of this paragraph should appear in the Methods section.

Reply: We kept the first sentence of the results because this is part of our results to mention that we successfully fit three models on the growth data (n=961) of boys and girls separately.

Comment: Again, I am surprised that no descriptive data of the population under study are presented. Growth data are considered without any health/clinical or socio-economic context, which limits the use, comparison of these results for other studies and the generalizability of the results. I think that minimal information should be presented.

Reply: We provided some available descriptive characteristics of the population in the supplementary table 2. We also added in the discussion:

“Another limitation of this study is generalizability of the results. These children lived in a semi-rural setting with a high prevalence of undernutrition. They also presented a high prevalence of potential risk factors for altered growth [4] (Supplementary Table 2). It is therefore possible that their growth would differ from a setting with different characteristics”.

Comment: Page 9, line 37: Figure 1 and figure???

Reply: Corrections made in the manuscript
“(Figure 1 and Figure 2)”

Comment: Page 10, line 26-28: where THEY WERE not measured

Reply: Corrections made in the manuscript:

“Using the Jenss-Bayley model, weight and height were calculated for all the children at different age points particularly when they were not measured”

Comment: Table 3, title: ESTIMATED WEIGHT AND weight growth velocity(Jenss-Bayley model)

Reply: Corrections made in the manuscript (now Table 4):

Table 4 Estimated weight and weight growth velocities (SD) of girls and boys aged 0 to 6 years, from the Jenss-Bayley model

Comment: Table 3, columns title should be: “Weight” and “Weight velocity” (“Estimated” is in the title and both are estimated)

Reply: Corrections made in the manuscript (now Table 4)

Comment: Same remarks in Table 4.

Reply: Suggested corrections made in the manuscript (now Table 5):

Table 5 Estimated length/height and length/height growth velocity (SD) of girls and boys aged 0 to 6 years, from the Jenss-Bayley model

Comment: The commentary Page 11, lines 21-22 is confusing. I understand that as the variance of the parameters is low compared to their estimation, the mean trajectory is reliable, but I don't understand why it suggests that “all the parameters highly contributed to the mean trajectory”. Parameters of the growth curves correspond to specific phases of growth and if some specific parameters had larger variances, one could comment on the phases of the mean growth that are less reliable than others. The sentence is too allusive. Authors should mention which conclusions they draw from that and why. And this should be placed in the Discussion section and not in the Results.

Reply: As the reviewer also highlighted, the conclusion was that all the fixed-effect parameters were reliable as they did not have large variances. However, we feel that this sentence was redundant so we removed it.

Comment: Comparison of models was made only on RSD, AIC, BIC and log-likelihood statistics. These are the relevant indicators to use. But they are not very concrete for readers. It may interesting to produce differences observed at some particular ages (3 months, 1 year, 3 years, 6 years, for example), in order to measure the size of the differences between the different models.

Reply: The reviewer mentioned that the presented fit indicators (RSD, AIC, BIC) may not be very concrete for readers. We took into account this feedback and presented delta AIC values (Table 2 in the manuscript) in order to quantify the size of the difference in AIC for the best fit model versus other candidate models.

Table 3 Comparison of the goodness of fit of the three candidate models

The delta AIC [25] for a candidate model, is the difference between the AIC values of the best and the candidate models. This difference (delta AIC) is then used as follows to determine the quantity of

support for each candidate model. If the delta AIC is

- o < 2 : there is substantial evidence to support the candidate model (i.e., the candidate model is almost as best performing as the best model).

- o Between 4 and 7: considerably less support for the candidate model to be the best model.

- o > 10 : there is essentially no support for the candidate model to better than the best model

Therefore, the difference between the AIC values of the Jenss-Bayley model (best fit model) and other two candidate models demonstrated quantifiable evidence that the Jenss model was the best model [25]. We also added the following in the discussion:

“Also, the difference between the AIC (delta AIC) of the Jenss-Bayley model (best model) and other two candidate models demonstrated quantifiable evidence that the Jenss-Bayley model was consistently the best one fitting on weight and height both for girls and boys [25]”

As the reviewer mentioned about further analysis and show differences in predictions at some particular ages, we took this into account and show in the following tables the values of estimated weight and height by three different models at 3 months, 1 year, 3 years, 6 years. It appears there are not major differences in the estimation of growth at different ages by the three models, the main difference being observed at three months.

Table 2. Prediction of growth at 3 months by three models compared

Table 3. Prediction of growth at 1 year by three models compared

Table 4. Prediction of growth at 3 year by three models compared

Table 5. Prediction of growth at 6 years by three models compared

Our conclusion from the above estimations are that all 3 models predict growth quite well without considerable variations in predictions (highest differences being observed at earlier ages), however based on the goodness-of-fit indicators it is preferable to use the Jenss-Bayley model.

Comment: The reader may also be interested in the comparison of performance between the original Gompertz model and the corrected version.

Reply: The original Gompertz model as published by Benjamin Gompertz in 1825 (now cited in in the manuscript) was on human mortality and was long of interest only to actuaries until recently when it was started to be used by authors as a growth curve both for biological and for economic phenomena (please refer to paper by Winsor (1932)). In fact, it appears that Davidson (1928) first used the Gompertz curve to describe growth using body weight of cattle cited in Winsor (1932). So in summary we do not think that the original Gompertz curve will appropriately represent child growth when growth is in its linear shape which also explains why we adapted the Gompertz model and used it directly in the actual study.

DISCUSSION

Page 11, line 55 : nutritional status AT 6 YEARS.

Reply: Corrections made in the manuscript

Comment: I think that discussion of the existing literature should focus on studies conducted in Sub-Saharan Africa, in populations potentially affected by undernutrition, as this is the main value of this present study: how popular growth models - mainly used and developed in developed countries - can fit growth in specific populations with a high prevalence of impaired growth?

Reply: We modified the discussion and focused the discussion of existing literature on the studies conducted in sub-Saharan Africa:

We added “in SSA” at the end of the following line:

“This paper extends the previous limited studies that either fitted the Jenss-Bayley model or compared its goodness of fit with other growth models in childhood.”

We removed the following discussion about the study in American children:

“Berkey[5] reported that the Jenss-Bayley model fitted well as compared to the Count model on the length and weight data during the early childhood of American children”

Comment: Page 12, lines 40-42: Differences in growth and nutritional status of children living in urban or rural settings are suggested. In which direction?

Reply: The initial idea was that to simply mention that differences in the nutritional status of children exist in both settings. As the reviewer suggested that mentioning that a difference exists may not be very useful. We decided not to discuss further on the direction because this is outside scope of the paper. We therefore decided to remove the sentence.

Comment: Page 12, Line 45: A strength of the study is that standardized measurements of weight and height were available (not only their prospective collection)

Reply: Corrections made in the manuscript:

“Third, the growth data used in this study were taken from a prospective study and were standardized measurements of weight and height; therefore, it is likely to be more accurate than data from routine surveys or health records”

Comment: Page 12, Line 47: Regarding the number of measurements, see my previous remark in the Method section.

Reply: We now describe the selection process of the population in the manuscript (response in the above comment). In addition, we also presented the characteristics of the children.

Comment: Page 13, lines 6-10: I don't understand what the authors mean when saying “On the other hand, reduced measurement errors ... may decrease residuals ...” ! Please clarify.

Reply: We removed this sentence to be clearer. It was intended to emphasize that if the measures are more accurate, this excludes a source of variability which could impact the quality of the modelling.

Comment: To improve the generalizability of the results of this study, models may be further compared regarding the interpretability of their parameters, and computational, parameterization and convergence issues. The models differ by the number of their parameters and the specific phases of growth that can describe. Depending on the future use of such models to study growth determinants, some of them may be more relevant, at comparable level of fit.

Reply: Regarding the generalizability of the study, we also replied to the first reviewer: “The sampling was not representative of the Beninese population. Our sample comes from an HIV-negative population in a rural setting where many health problems (e.g. undernutrition, infectious diseases) are common. Therefore, we acknowledge that the generalizability of our results is limited. Other studies using subjects from settings other than this region (for example from urban setting) will be useful to compare the consistency of the results. The idea behind this study was not generalizability but rather finding the best model among potentially good models in this particular population of children with a high prevalence of malnutrition and to study a new model (adapted Gompertz model).”

We also added in the discussion:

“Another limitation of this study is generalizability of the results. These children lived in a semi-rural setting with a high prevalence of undernutrition. They also presented a high prevalence of potential risk factors for altered growth [4] (Supplementary Table 2). It is therefore possible that their growth would differ from a setting with different characteristics”.

Furthermore, the models do not differ in the number of their parameters. All 3 models compared in this study have four parameters.

We added the following paragraph in the comparison of the Goodness of Fit in the Methodology section:

“The delta AIC [25] for a candidate model, is the difference between the AIC values of the best and the candidate models. This difference (delta AIC) is then used as follows to determine the amount of support for each candidate model. If the delta AIC is < 2 , there is substantial evidence to support the candidate model (i.e., the candidate model is almost as best performing as the best model). If the delta AIC is between 4 and 7, there is considerably less support for the candidate model to be the

best model. Finally, if the delta AIC is >10 , there is essentially no support for the candidate model to be better than the best model”

We added the following paragraphs in the RESULTS section regarding the Goodness of fit for weight and height models:

“For weight models, the Jenss-Bayley model had the lowest AIC and BIC values for the weight models, both in boys and girls, indicating a better fit than the two other candidate models (Table 3). Both the Reed and the adapted Gompertz model had high delta AIC values for both boys (136 and 117, respectively) and girls (129 and 135, respectively) indicating essentially no support for these two candidate model to be better than the best fitting model (Jenss-Bayley).

“For height models, the Jenss-Bayley model and the adapted Gompertz model had the lowest AIC and BIC values for the height model in boys while the Jenss-Bayley model had the lowest AIC values in girls than the Reed and adapted Gompertz model. With regard to delta AIC values, both the Reed and the adapted Gompertz model had values less than 10 for boys (8 and 6, respectively) indicating considerably less support for these two models to be the best fitting model. While in girls, these two candidate models had delta AIC values >10 (28 and 32, respectively) which does not indicate any support for these two models to have better fit than the Jenss-Bayley model.”

We added the following paragraph in the DISCUSSION section to discuss models further in terms of fit and predictions:

“In general, the three models (Jenss-Bayley, adapted Gompertz and the Reed model) seemed to fit well both on weight and height data as was evidenced by mean residuals close to zero (Figure 1 and Figure 2)”

“Further analysis was done to show whether predictions by the three models (Jenss-Bayley, Reed and adapted Gompertz models) at different age points i.e. at 3 months, 1 year, 3 years, and 6 years were different. The predictions differ mainly at 3 months with no major differences afterward and the goodness of fit indicators supported the use of Jenss-Bayley model”.

In terms of computational and convergence of models, we added in the discussion:

“There were slight variations in the way the models converged. The Jenss-Bayley model converged easily on weight data using an unstructured variance-covariance matrix for random effects. More convergence issues were faced after attempting to converge the Jenss-Bayley model on the height data when the number of iterations was extended to 20000. The Reed model and the adapted Gompertz model also had convergence issues both on weight and height data with an unstructured variance-covariance matrix but were able to converge with a diagonal matrix.

Although an unstructured variance-covariance matrix would have been preferred for all the models, there was no difference in the fit indicators (i.e. AIC/BIC and RSD), fixed effects and predictions? (data not shown) when using a diagonal variance-covariance matrix on the Jenss-Bayley model. In particular, a diagonal variance-covariance matrix for random effects does not constraint the growth parameters to be uncorrelated, the four model parameters being free to vary for each individual.

Convergence issues could be affected by several reasons including but not limited to the number of measurements occasions and how the time intervals between measurement occasions are spaced, complexity of the model (e.g. number of parameters, monotonicity), parameterization (e.g. addition of higher order terms such as $\ln(\text{age})$ [14], as well as the type of statistical packages. Simpler methods than SAEMIX could have been used, but for the purpose of consistency in methods it was preferred to fit them with the same one. Some computational and convergence issues we faced could be explained by the limited measurement occasions and unequally spaced time intervals between 2 and 6 years as also reported by a previous study in South-African [14]. To facilitate the convergence of the models, constraints of positivity on the parameters of all three models was applied by using exponential functions, as shown in equation 2 for the Jenss-Bayley model[1]”

In terms of parameterization and the description and interpretation of growth models can be referred

to the METHODS section under “Growth Models”.

CONCLUSION

Last sentence: PARAMETERS of this model

Reply: Corrections made in the manuscript

Other amendments in the manuscript:

1. We added “Growth trajectory” to the keywords

2. We added following in the discussion:

“The mean height growth trajectories were relatively parallel with no visible difference in the direction of the curve between boys and girls. On the contrary visible difference in direction was observable in the mean weight growth trajectory of boys and girls. This difference was visible after infancy where the trajectory of girls falls below the curve of boys.”

3. We also made a correction to a typo error in the Table 5 in the manuscript:

In the column of girls predicted length, height: the first value 48.63 (2.29) was replaced by 58.60 (1.42)

4. Since a new table has been added in the manuscript, tables have been renumbered:

Table 1. Summary of some studies that have used quadratic, structural and non-structural models to describe physical growth

Table 2 Number of anthropometric measures per child and per sex

Table 3 Comparison of the goodness of fit of the three candidate models

(In table 3, the full name of the Jenss model has been added i.e. Jenss-Bayley model)

Table 4 Estimated weight and weight growth velocities (SD) of girls and boys aged 0 to 6 years, from the Jenss-Bayley model

Table 5 Estimated length/height and length/height growth velocity (SD) of girls and boys aged 0 to 6 years, from the Jenss-Bayley model

We rounded off some values values in table 4 and table 5.

5. Supplementary tables have been reviewed. They now follow the following order:

Supplementary Table 1. Number of children with anthropometric measurements at different age intervals. Data from boys (n=461) and girls (n=500) from birth to around six years of age

Supplementary Table 2. Summary of maternal and child characteristics in the prospective child cohort in Benin, Sub-Saharan Africa.

Supplementary Table 3. Parameters estimates for three candidate models a fitted to weight and height

Supplementary Table 4. Stunting, underweight and wasting in children using predicted data from the Jenss-Bayley model. n (%)

6. Correction in the supplementary Table 4: we made a correction by providing denominators and a minor correction to the percentages of wasting.

Supplementary Table 4. Stunting, Underweight, Wasting in children using predicted data from the Jenss-Bayley model. n (%)

References

1. Botton, J., et al., Postnatal weight and height growth modeling and prediction of body mass index as a function of time for the study of growth determinants. *Ann Nutr Metab*, 2014. 65(2-3): p. 156-66.

2. Comets, E., A. Lavenu, and M. Lavielle, Parameter Estimation in Nonlinear Mixed Effect Models Using saemix, an R Implementation of the SAEM Algorithm. *Journal of Statistical Software*, 2017.

80(3): p. 1-42.

3. Burnham, K.P. and D.R. Anderson, Model selection and multimodel inference. A practical information-theoretic approach 2ed. 2002: Springer.
4. Accrombessi, M., et al., Malaria in Pregnancy Is a Predictor of Infant Haemoglobin Concentrations during the First Year of Life in Benin, West Africa. PLoS One, 2015. 10(6): p. e0129510.
5. Berkey, C.S., Comparison of two longitudinal growth models for preschool children. Biometrics, 1982. 38(1): p. 221-34.
6. Hauspie, R.C., Mathematical models for the study of individual growth patterns. Rev Epidemiol Sante Publique, 1989. 37(5-6): p. 461-76.
7. Scherdel, P., et al., Growth monitoring as an early detection tool: a systematic review. . The Lancet Diabetes & Endocrinology 2016. 5(4): p. 447-56.
8. Kramer, M.S., et al., Growth During Infancy and Early Childhood and Its Association with Metabolic Risk Biomarkers at 11.5 Years. Am J Epidemiol, 2019.
9. Regnault, N. and M.W. Gillman, Importance of characterizing growth trajectories. Ann Nutr Metab, 2014. 65(2-3): p. 110-3.
10. Botton, J., et al., Postnatal weight and height growth velocities at different ages between birth and 5 y and body composition in adolescent boys and girls. Am J Clin Nutr, 2008. 87(6): p. 1760-8.
11. Molinari, L. and T. Gasser, The human growth curve: distance, velocity and acceleration. In: Hauspie, R C; Cameron, N; Molinari, L. Methods in Human Growth Research. 2004, Cambridge University Press: Cambridge UK. p. 27-54.
12. Hauspie, R.C., Cameron, N. & Molinari, L. (eds.), Methods in Human Growth Research. 2004: Cambridge Press.
13. Olusanya, B.O. and J.K. Renner, Predictors of growth velocity in early infancy in a resource-poor setting. Early Hum Dev, 2011. 87(10): p. 647-52.
14. Chirwa, E.D., et al., Multi-level modelling of longitudinal child growth data from the Birth-to-Twenty Cohort: a comparison of growth models. Ann Hum Biol, 2014. 41(2): p. 168-79.
15. Singer, J.D. and J.B. Willett, Applied longitudinal data analysis: modeling change and event occurrence. 2003, New York: Oxford University Press
16. Cameron, N., et al., Timing and magnitude of adolescent growth in height and weight in Cape coloured children after kwashiorkor. J Pediatr, 1986. 109(3): p. 548-55.
17. Simondon, K.B., et al., Comparative study of five growth models applied to weight data from congolese infants between birth and 13 months of age. Am J Hum Biol, 1992. 4(3): p. 327-335.
18. Pagezy, H. and R. Hauspie, Growth in weight of African babies, aged 0–24 months, living in a rural area at the Lake Tumba, Zaire. Annals of Tropical Paediatrics, 1985. 5(1): p. 41-47.
19. Rozzi, F.V., et al., Growth pattern from birth to adulthood in African pygmies of known age. Nat Commun, 2015. 6: p. 7672.
20. Padonou, G., et al., Factors associated with growth patterns from birth to 18 months in a Beninese cohort of children. Acta Trop, 2014. 135: p. 1-9.
21. Gonzalez, R., et al., Intermittent preventive treatment of malaria in pregnancy with mefloquine in HIV-negative women: a multicentre randomized controlled trial. PLoS Med, 2014. 11(9): p. e1001733.
22. Ahmadi, S., et al., Hunting, Sale, and Consumption of Bushmeat Killed by Lead-Based Ammunition in Benin. Int J Environ Res Public Health, 2018. 15(6).
23. Bodeau-Livinec, F., et al., Neurocognitive testing in West African children 3-6 years of age: Challenges and implications for data analyses. Brain Research Bulletin, 2018.
24. Littell, R.C., P. Jane, and N. Ranjini, Modelling covariance structure in the analysis of repeated measures data. Statistics in Medicine, 2000. 13(19): p. 1793-1819.
25. Anderson, C., et al., Comparing predictive abilities of longitudinal child growth models. Stat Med, 2018.

REVIEWER	Sara Benjamin-Neelon Johns Hopkins University USA
REVIEW RETURNED	22-May-2020

GENERAL COMMENTS	Thank you for the opportunity to review this revised manuscript. The authors have done a nice job responding to the reviewer comments. One minor issue remains. 1. The inclusion of the new Table 1 is somewhat confusing. Why include this information in a table? The title ("Summary of some studies...") implies that the authors did not include all studies but instead selected some studies. But, it does not seem appropriate to include a table summarizing prior research for an article that is not a systematic or narrative review. It would be more helpful to summarize these prior studies briefly in the text rather than in a table.
---

REVIEWER	Izzuddin M Aris Harvard Medical School and Harvard Pilgrim Health Care Institute
REVIEW RETURNED	10-Apr-2020

GENERAL COMMENTS	The authors have addressed my concerns and queries adequately. I have no further comments.
--

REVIEWER	Nathalie Costet Univ Rennes, Inserm, EHESP, Irset (Institut de recherche en santé, environnement et travail) - UMR_S 1085, F-35000 Rennes, France
REVIEW RETURNED	23-Apr-2020

GENERAL COMMENTS	The authors proposed a substantially improved version of their manuscript. They considered all the issues raised by the different reviewers. In particular, the context of the study, the selection and the description of the population, the literature review are now well presented. Technical details about the growth models are given and will be useful for comparison with earlier and future studies. I found some minor typographical or editorial errors that may be corrected :  - P6, line10 : "and. Besides, the parameters" - P7, line 53 : "at mean 6 years of age" sounds strange. "at 6 years on average" ? - P8, line 48 : idem - P9, line 48 : delete "it" before "centring" and replace "centring" by "centering", and "centred" by "centered" - P10, Equation 2 : the minus (-) character before B_i should be superscripted - P20, line 52 : add "(data not shown) " at the end of this sentence - P21, line 48 : WERE applied Finally, P21, line 22: in order to be more explicit about possible correlations between individual growth model parameters, despite a diagonal matrix between the random effects, I would say "as the fixed part of the individual growth parameters (population parameters) are free to be correlated anyway".
---

VERSION 2 – AUTHOR RESPONSE

Reviewer: 1

The first reviewer, Sara Benjamin-Neelon, wrote:

Thank you for the opportunity to review this revised manuscript. The authors have done a nice job responding to the reviewer comments. One minor issue remains.

1. The inclusion of the new Table 1 is somewhat confusing. Why include this information in a table? The title ("Summary of some studies...") implies that the authors did not include all studies but instead selected some studies. But, it does not seem appropriate to include a table summarizing prior research for an article that is not a systematic or narrative review. It would be more helpful to summarize these prior studies briefly in the text rather than in a table.

Reply: Thank you for pointing this out. As the reviewer suggested we have now removed the table 1 and have kept the citation of these studies in the introduction.

"Many studies have analysed child growth data in Africa, but few studies[6, 7, 13, 14, 16-18] have applied structural or non-structural growth models on African longitudinal growth data. Very few of these studies[7, 14] have assessed the comparative merits of models like the Jenss-Bayley and Gompertz model on child growth in SSA."

We avoided to discuss in detail these studies in the introduction as some of these studies (Pagazey and Hauspie (1985); Simondon et al. (1992); Chirwa et al. (2014)) are already discussed in the Discussion section:

"This paper extends the previous limited studies that either fitted the Jenss-Bayley model or compared its goodness of fit with other growth models in childhood in SSA. Pagazey and Hauspie[6] successfully fitted the Jenss-Bayley model on Congolese babies weight data. However, the phase of growth studied in this study was restricted around birth to two years. Similarly, few studies reported Reed model fitted well. Simondon and colleagues[7] compared several models on the growth data of 95 Congolese children and reported that the Reed model had a better fit but this study did not include the Jenss-Bayley and the adapted Gompertz models and studied child growth only during infancy. Chirwa and colleagues[14] also reported that the Reed model fitted well on the growth data from birth to 10 years in 453 children living in an urban setting in South Africa. The study reported that the Jenss-Bayley model did not fit well in the early years. However, this study included children only from an urban setting and did not test the performance of the Gompertz model."

Reviewer: 2

The second reviewer, Izzuddin Aris, wrote:

The authors have addressed my concerns and queries adequately. I have no further comments.

Reply: Thank you for considering our responses.

Reviewer: 3

The third reviewer, Nathalie Costet, wrote:

« The authors proposed a substantially improved version of their manuscript. They considered all the issues raised by the different reviewers. In particular, the context of the study, the selection and the description of the population, the literature review are now well presented. Technical details about the growth models are given and will be useful for comparison with earlier and future studies.

Reply: Thank you for accepting our responses.

Comment 1 :

I found some minor typographical or editorial errors that may be corrected :

- P6, line10 : "and. Besides, the parameters"

Reply: We corrected in the manuscript:

"In contrast, non-structural models do not formulate any particular form of growth curve and may demonstrate instability at extremities. Besides, the parameters obtained from non-structural models do not provide any biological interpretation[12, 15]."

Comment 2 : - P7, line 53 : "at mean 6 years of age" sounds strange. "at 6 years on average" ?

Reply: We corrected in the manuscript:

"and finally assessed in the EXPLORE study at 6 years on average".

Comment 3 : P8, line 48 : idem

Reply: We corrected in the manuscript:

« Finally, at age 6, children were assessed within the EXPLORE study at 6 years on average.»

Comment 4 : P9, line 48 : delete "it" before "centring" and replace "centring" by "centering", and "centred" by "centered"

Reply: We corrected in the manuscript but we kept the word "centring" and "centred" which is in line with the use of British English spelling in the manuscript.

Comment 4 : P10, Equation 2 : the minus (-) character before Bi should be superscripted

Reply: We corrected in the manuscript:

$$E(y_{ij}) = \exp A_i + \exp -B_i \cdot t_{ij} + \exp C_i \cdot (1 - \exp -\exp -D_i \cdot t_{ij}) \quad (2)$$

Comment 5 : P20, line 52 : add "(data not shown) " at the end of this sentence

Reply: We corrected in the manuscript:

"The predictions differ mainly at 3 months with no major differences afterwards and the goodness of fit indicators supported the use of Jentsch-Bayley model (data not shown)."

Comment 5: P21, line 48: WERE applied

Reply: We now corrected in the manuscript:

"To facilitate the convergence of the models, constraints of positivity on the parameters of all three models were applied by using exponential functions, as shown in equation 2 for the Jentsch-Bayley model[28]"

Comment 6: Finally, P21, line 22: in order to be more explicit about possible correlations between individual growth model parameters, despite a diagonal matrix between the random effects, I would say "as the fixed part of the individual growth parameters (population parameters) are free to be correlated anyway".

Reply: We corrected the following paragraph:

"In particular, a diagonal variance-covariance matrix for random effects does not constraint the growth parameters to be uncorrelated, the four model parameters being free to vary for each individual."

and replaced with the following paragraph:

« In particular, a diagonal variance-covariance matrix for random effects does not constraint the growth parameters to be uncorrelated, as the fixed part (population parameters) are free to be correlated anyway »

Other amendments/edits in the manuscript:

The affiliation of the 3rd co-author have been corrected to « Institut de Recherche pour le Développement (IRD), Cotonou, Benin »

Following minor edits have been made in the manuscript.

1. In the Abstract, under the section “Participants” we replaced “from birth to six years of age” to “aged 0-6 years” to reduce words.

2. In the Abstract, under the results section:

a. “The Jenss-Bayley model presented the best fit for weight and height, both in boys and girls”, we removed “for” before the “height” as we found it redundant.

b. In the sentence “while mean weight growth curve of girls fell slightly below the curve of boys after neonatal life”, we added an article “the” before “mean weight growth curve”

3. In the Introduction,

a. we removed “on child growth in SSA” just after “Jenss-Bayley” and kept “on child growth in SSA” just after “Gompertz model” as it was a repetition/redundancy:

“Very few of these studies[7, 14] have assessed the comparative merits of models like the Jenss-Bayley and Gompertz model on child growth in SSA. Furthermore, as to our knowledge, no study has compared the fitness of the adapted Gompertz growth model on child growth data.”

b. In the last paragraph of Introduction: « between birth and 6 years of age » is replaced with “aged 0-6 years”

4. Methodology :

a. Under the section “Study design and data sources”:

i. “1 month, and also at 9 and 12 months of age” is replaced with “1 month, and also 9 and 12 months old”.

ii. “The TOVI study followed-up at 1 year-of-age 747 singleton children” is replaced with “The TOVI study followed-up 747 singleton 1-year-old children”

iii. “92 children randomly selected were assessed between 3 and 5 years” is replaced with “92 children randomly selected were assessed between 3 and 5 years”

b. In the « Comparison of Goodness of Fit », we added “be” before the “better than the best model”:
“Finally, if the delta AIC is >10 , there is essentially no support for the candidate model to be better than the best model”

5. In the Results section, the numbering of the tables is changed due to the removal of Table 1 (suggested by reviewer 1)

Table 1 Number of anthropometric measures per child and sex

Table 2 Comparison of the goodness of fit of the three candidate models

Table 3 Estimated weight and weight growth velocities (SD) of girls and boys aged 0 to 6 years, from the Jenss-Bayley model

Table 4 Estimated length/height and length/height growth velocity (SD) of girls and boys aged 0 to 6 years, from the Jenss-Bayley model

6. At the beginning of the Results section, “height” is added before “growth data of 961 children from birth to 6 years”:

“Three models were fitted to weight and height growth data of 961 children from birth to 6 years..”

7. In the Discussion, paragraph 2, we added « also » after « Chirwa and colleagues[14] »

« Chirwa and colleagues[14] also reported that the Reed model fitted well... ».

Also, we added « only » before « from urban setting » referring to the study by Chirwa and colleagues (2014) :

“However, this study included children only from an urban setting...”

8. In the entire manuscript, numbers less than ten are now written out in full, except age, percentages and when fractions or decimals are involved. Additionally, “x years of age” (where x is a given age) have been replaced by “age x” or “x years” to avoid tautology. E.g. in the abstract « 4% of girls were estimated to be underweight, wasted, and stunted at 6 years of age » have been replaced with « 4% of girls were estimated to be underweight, wasted, and stunted at 6 years »